# Intravenous Morphine Infusion versus Thoracic Epidural Infusion of Ropivacaine with Fentanyl after the Ravitch Procedure—A Single-Center Cohort Study

**DOI:** 10.3390/ijerph191811291

**Published:** 2022-09-08

**Authors:** Dariusz Fenikowski, Lucyna Tomaszek

**Affiliations:** 1Department of Thoracic Surgery, Institute of Tuberculosis and Lung Diseases, Rabka-Zdrój Branch, 34-700 Rabka-Zdrój, Poland; 2Faculty of Medicine and Health Sciences, Andrzej Frycz Modrzewski Krakow University, 30-705 Kraków, Poland

**Keywords:** Ravitch procedure, intravenous morphine, thoracic epidural, postoperative pain assessment, anxiety, patients satisfaction, nursing care

## Abstract

Objective. To compare the efficacy of analgesia with intravenous infusion of morphine and thoracic epidural infusion of ropivacaine with fentanyl in pediatric patients after the Ravitch procedure. Methods. Postoperative analgesia was achieved by intravenous morphine infusion with a dose of 0.02–0.06 mg/kg per hour (intravenous group, *n* = 56) or thoracic epidural infusion of 0.2% ropivacaine and fentanyl 5 µg/mL with a flow rate of 0.1 mL/kg per hour (epidural group, *n* = 40). Furthermore, the multimodal pain management protocol included paracetamol, non-steroidal anti-inflammatory drugs, and metamizole as a rescue drug. The primary outcomes included pain scores (according to the Numerical Rating Scale, range 0–10), while the secondary outcomes included consumption of the rescue drug, anxiety, postoperative side effects, and patient satisfaction. The observation period lasted from postoperative day 0 to postoperative day 3. Results. Median average and maximal pain scores at rest, during deep breathing, and coughing were significantly lower in the intravenous group compared to the epidural group (*p* < 0.05). The effect size was medium (Cohen’s d ranged from 0.5 to 0.7). Patients receiving morphine required significantly lower numbers of metamizole doses than in the epidural group (median 1 vs. 3; *p* = 0.003; Cohen’s d = 0.6). Anxiety, postoperative side effects, and patient satisfaction were similar in both groups (*p* > 0.05). Conclusions. An intravenous infusion of morphine may offer better postoperative analgesia than a thoracic epidural infusion of ropivacaine with fentanyl.

## 1. Introduction

Pediatric thoracic surgery also focuses on treating patients with malformations of the thoracic wall, such as pectus excavatum, pectus carinatum, or mixed excavatum-carinatum deformity. Anomalies include both the sternum and the adjacent chondrocostal structures and are more prevalent in males. Pectus excavatum, the most common chest-wall deformity, is a depression of the sternum, while pectus carinatum, considered the second-most common cause of thoracic malformations, is a protrusion of the sternum [1]. One of the surgical techniques used for the treatment of chest wall abnormalities is the modified Ravitch procedure according to Buchwald [2]. It is a one-time, conclusive corrective surgery with no necessity of applying extra brackets to restore the anatomical shape of the thoracic wall, with favorable long-term outcomes.

This modified Ravitch procedure is an open surgery normally performed during adolescence. Postoperative pain originates from surgical damage to the skin, muscles (pectoral major, rectus abdominis), and the sternum or ribs (costal cartilages). Pain also comes from the presence of a chest tube in the lower pole of the surgical wound or pleural cavity (in the case of pneumothorax) [2]. The severity of pain increases with deep breathing, coughing, and intense body movements [3]. Poor pain management may result in harmful clinical (e.g., insufficient mobilization and deep breathing lead to pneumonia), psychological (e.g., anxiety), and socioeconomic consequences (e.g., decreased patient satisfaction, increased length of stay, increased cost of care) [4,5]. Therefore, good pain management is a key component of recovery after surgery. 

Practical guidelines for perioperative analgesia in the pediatric population [6] recommend the multimodal approach, which is defined as the use of two or more analgesic agents that target different receptors in the pain pathway. The goal of this pain relief strategy is to improve analgesia without increasing the side effects compared with increased doses of single agents [7]. A multimodal analgesic protocol may contain both systemic analgesics and interventional techniques [8]. Typical elements of systemic analgesic options in pediatric patients undergoing the Ravitch procedure are usually opioids, non-steroidal anti-inflammatory drugs, paracetamol, diazepam, or gabapentin [9,10], while continuous epidural anesthesia appears to be one of the most commonly used interventional techniques [11,12]. Recent studies have also shown the benefit of paravertebral and intercostal blocks [13] or cryoanalgesia [10,14] in pain control after chest wall reconstruction using the Ravitch procedure.

There is evidence that epidural analgesia provides superior pain relief compared to intravenous analgesia [15]. However, multimodal protocols, an essential part of which are opioids given by intravenous route, may be as effective as those containing local anesthetics delivered through epidural catheters [16,17]. Unfortunately, both methods are not free of complications, such as postoperative nausea and vomiting, urinary retention, pruritus, and respiratory depression [18,19]. Additionally, in the case of epidural analgesia, there is a low risk of serious neurological adverse events (e.g., permanent nerve injury, epidural abscess) and local anesthetic systemic toxicity [18,20].

Typically, the choice of a management strategy depends on patient preference and the experience of the physician. There is no consensus as to an ideal analgesic strategy. Thus, the current study aimed to compare the efficacy of nurse-controlled analgesia with intravenous infusion of morphine and epidural infusion of ropivacaine with fentanyl in pediatric patients undergoing the Ravitch procedure.

## 2. Methods

### 2.1. Trial Design, Setting

This was a single-center cohort study conducted at the Department of Thoracic Surgery of the Institute of Tuberculosis and Lung Disease, Rabka Zdroj Branch, from May 2017 to December 2020. The study was prospectively registered at ClinicalTrails.gov (ID: NCT03393702) as randomized, blinded, placebo-controlled to investigate the usefulness of gabapentin as a component of a multimodal analgesic regimen in the postoperative period in pediatric patients whose pain was treated via an intravenous and epidural route. The study protocol was in compliance with the Declaration of Helsinki and was approved by the Local Bioethics Committee at the National Institute of Tuberculosis and Lung Diseases in Warsaw (decision numbers: KB-6/2017; KB-125/2019).

### 2.2. Participants

All patients 9–17 years of age who were to undergo a scheduled Ravitch procedure, with postoperative chest drainage, in whom postoperative analgesia with intravenous morphine or epidural analgesia with ropivacaine/fentanyl was used and whose parents signed informed consent were included. Consent was also obtained from patients 16 years of age. Exclusion criteria were: American Society of Anesthesiologists (ASA) physical classification ≥3, postoperative intubation, oncological patients, inability to communicate pain, known drug allergy to local anesthetics or opioids, history of chronic pain or preoperative analgesic use, those patients suffering from mental illnesses or epilepsy.

### 2.3. Interventions

#### 2.3.1. Drugs

The multimodal protocol for the perioperative and intraoperative administration of drugs in the intravenous and epidural groups is shown in Table 1. All patients received the same premedication, antiemetic prophylaxis and treatment, pre-emptive analgesia, induction and maintenance of anesthesia, and non-opioid postoperative analgesia. Perioperative gabapentin was received by 50% of the patients in each group according to the following scheme: 1 h before surgery at a dose of 15 mg/kg, and after surgery at a dose of 7.5 mg/kg two times per day (at 6:00 a.m. and 6:00 p.m.) for 3 days. Intraoperative analgesia in the intravenous group was provided by fentanyl administered every 20–30 min, while in the epidural group, patients were given local anesthetics. Postoperative analgesia was achieved via intravenous morphine infusion (with a dose of 0.02–0.06 mg/kg per hour) or epidural infusion of 0.2% ropivacaine and fentanyl 5 µg/mL (with a flow rate of 0.1 mL/kg per hour), respectively.

#### 2.3.2. Epidural Catheter

An epidural catheter was used in the anesthetized patients in the lateral decubitus position at the thoracic levels—a T4–T7 sensory level is appropriate for the Ravitch procedure. Using a median approach, the epidural space was identified by loss of resistance to air with an 18-G or 20-G Tuohy needle (children up to 10 years of age). A 20-gauge or 24-gauge catheter, respectively, was inserted through the needle up to 3–4 cm in the cephalic direction and tunneled laterally from the initial puncture. Next, the place of catheter insertion was secured with a sterile transparent dressing. To protect against infective material anti-bacterial filter was inserted at the junction of the epidural catheter and infusion line. Furthermore, the epidural infusion was labeled ‘Epidural,’ and the label, epidural catheter, syringe, and tubes for infusion lines were yellow. 

#### 2.3.3. Patient Monitoring by Nurses

Patients after surgery were hospitalized in the intensive care unit for three postoperative days at a minimum. Immediately upon admission to the ward, the Simplified Post-Anesthetic Recovery Score scale, ranging from 0 to 6, as described by Steward, was used. Fully recovered patients received 6 points (they were awake, could cough on command or cry, and could move their limbs purposefully) [21].

During the observation period, blood pressure, heart rate, respiratory rate, oxygen saturation, and temperature were continuously monitored. Pain scores and sedation scores were measured at the same time intervals as the degree of motor block in the epidural group (on the first postoperative day: for the first 4 h, every hour, then at least every 4 h; on the second and third postoperative days: at least four times a day; 30 min after additional analgesic administration). 

The pain was scored using the Numeric Rating Scale (NRS) according to three steps: at rest, during deep breathing, and when coughing (Figure 1). Scores ranged from 0 to 10—higher scores depict higher pain intensity. Pain assessment helps to identify inadequate analgesia (NRS ≥ 3). Both mean average and maximum pain scores were calculated for each patient (at rest, during deep breathing, and when coughing) over the postoperative period (days 0–3).

Sedation was assessed based on the motor responsiveness of the patient using a modified Ramsay’s Sedation Scale. This scale uses a numeric score from 1 to 5 (score 1: anxious or agitated; score 2: cooperative and oriented and tranquil; score 3: asleep, easy to wake; score 4: asleep, difficult to wake; score 5: asleep, does not respond to a painful stimulus). Monitoring the level of sedation helps to detect opioid-induced respiratory depression (scores 4 or 5 = oversedation). 

The motor blockage was graded using the Bromage scale (grade 0: no paralysis, full flexion of knees and feet; grade 1: partial, just able to move knees; grade 2: almost complete, able to move feet only; grade 3: complete, unable to move knees or feet) [22]. The increasing degree of motor block may indicate excessive epidural drug administration or catheter penetration into the subarachnoid space or the development of an epidural hematoma or abscess.

The state and trait components of anxiety were measured by the State-Trait Anxiety Inventory for patients between 9 and 14 years of age (STAI-C) [23] and older (STAI) [24]. The state anxiety (i.e., how one feels at the moment) and trait anxiety questionnaires (i.e., how one generally feels) consist of 20 items to which the patient responds by selecting one of four categorized responses. The final results were shown as sten scores with a range 1–10 (moderate anxiety: 5–6 sten, high anxiety: ≥7 sten). State anxiety was assessed one day prior to surgery (together with trait anxiety) and on postoperative day 3 (together with the patient’s satisfaction).

Satisfaction from patient postoperative pain management was assessed using the NRS scale four times: on the day of surgery, on postoperative day 1, postoperative day 2, and postoperative day 3. Scores ranged from 0 (completely dissatisfied) to 10 (completely satisfied), with the higher scores indicating greater patient satisfaction. The mean average score was calculated for each patient over postoperative days 0–3.

#### 2.3.4. Nurse-Controlled Analgesia

A solution of morphine or ropivacaine with fentanyl was prepared and administered using programmable infusion pumps by trained nursing staff—only anesthesiology nurses were authorized to administer drugs via the epidural route. The infusion was maintained until the chest tube was removed.

In the case of inadequate pain relief (NRS ≥ 3), analgesia was modified by increasing the flow rate of the infusion of morphine or ropivacaine with fentanyl by 10–30% and/or administration of these analgesics in a bolus (half-hourly dose). Intravenous metamizole was given as a rescue drug.

### 2.4. Outcomes

The primary outcome measurements were average and maximal pain scores. Secondary outcome measures included the number of doses of metamizole, anxiety, side effects profile of postoperative analgesia, and patient satisfaction.

### 2.5. Statistical Analysis

Categorical variables were summarized as absolute numbers and percentages, and continuous variables as median, lower and upper quartile. The analysis of the existence of an association between categorical variables was performed with the Chi-square test or the Fisher exact test. The Mann–Whitney test was used to analyze the continuous variables (value of the Shapiro–Wilk test < 0.05). Statistical analysis was carried out using the Statistica 13 program (StatSoft^®^, Krakow, Poland). A *p*-value of 0.05 or lower was considered statistically significant.

The effect size was computed using a calculator for the test statistics of the Mann–Whitney-U [25]. It was shown as Cohen’s d and interpreted as a small, medium, or large effect (from 0.2 to <0.5, 0.5 to <0.8, and ≥0.8, respectively).

## 3. Results

### 3.1. Characteristics of Patients

One hundred twenty-five patients were evaluated for enrollment in the study. Data were obtained from 96 patients (76.8%) who fulfilled the inclusion and exclusion criteria. Fifty-six patients were in the intravenous, and 40 patients were in the epidural group (Figure 2).

Pectus excavatum and carinatum were the reasons for operation for 85.4% and 12.5% of patients, respectively (*n* = 82; *n* = 12), while mixed pectus deformities were recognized in only 2 patients (2.1%). Table 2 summarizes the demographic and clinical characteristics of the patients. Significant intergroup differences (*p* < 0.05) were observed regarding ASA, preoperative heart rate, and systolic/diastolic blood pressure, as well as the duration of surgery.

In the epidural group, 50% of the catheters (*n* = 20) were inserted at the Th6–7 nerve root level, 27.5% (*n* = 11) of the catheters at the Th5–6, 17.5% (*n* = 7) were inserted at the Th4–5, and two catheters (5%) below the Th7. The success rate was 60% (*n* = 24) after the first attempt, 27.5% (*n* = 11) of patients required two attempts, and in the remaining patients, three (*n* = 3), four (*n* = 1) and five (*n* = 1) attempts were made. A vascular puncture through a needle occurred in one patient. The paramedian approach was used for one patient.

In the postoperative period in all patients, the median heart rate was 77 [70; 84] beats/min, systolic and diastolic blood pressure was 109 [105; 114] mmHg and 61 [58; 64] mmHg, respectively, and oxygen saturation was 96% [95; 96]. Patients received oxygen through the nasal cannula with a flow of 1–2 L/min—median oxygen supplementation was 31 [19; 40] hours. There were no significant differences between the intravenous and epidural groups regarding the above-mentioned variables (*p* > 0.05).

### 3.2. Average and Maximum Postoperative Pain

The average median pain score, regardless of the measurement conditions, for all patients was <1/10, while the maximum pain at rest was 5/10, and during deep breathing/coughing, this was 3/10. Total postoperative average and maximum pain scores at rest, during deep breathing, and during coughing were significantly lower in the intravenous group compared to the epidural group. The effect size was medium (Cohen’s d ranged from 0.5 to 0.7) (Table 3).

It should be noted that no significant differences (*p* > 0.05) were observed in patient pain scores (average/maximum) between the groups only on the day of surgery and on the second postoperative day when the pain was measured during deep breathing and coughing.

### 3.3. Analgesic Consumption

Over the whole period of observation, the median consumption of morphine in the intravenous group was 20,000 [17,000; 23,000] μg, while the epidural group consumed 270 [204; 311] mL 0.2% ropivacaine and 1287 [987; 1532] μg fentanyl. There were no significant differences between the groups in number of hours of intravenous and epidural infusion (median 81 [75; 90] vs. 78 [69; 92]; Z = 0.39; *p* = 0.69), number of bolus (median 1 [0; 2] vs. 0.5 [0; 2]; Z = 0.32; *p* = 0.74) and number of increasing/decreasing infusion rate (median 5 [3.5; 6.5] vs. 5 [4; 6]; Z = 0.54; *p* = 0.59). In addition, on the third postoperative day, when the infusion of morphine and ropivacaine with fentanyl was stopped due to removal of the chest drainage, two patients in the intravenous group and four patients in the epidural group received tramadol.

In the epidural group, patients required a larger number of doses of metamizole than in the intravenous group: the median was 3 [1.5; 4] vs. 1 [1; 3]; Z = −2.93; *p* = 0.003. This effect size was medium (Cohen’s d = 0.6). 

### 3.4. Anxiety

No statistically significant differences were found between the intravenous and epidural groups regarding trait anxiety and preoperative/postoperative state anxiety (Table 4).

### 3.5. Side Effects

Side effects associated with postoperative analgesia are reported in Table 5. Although in the epidural group patients had more frequent nausea and vomiting compared to patients receiving intravenous morphine (*p* = 0.05), we did not find any significant differences in the median numbers of these side effects (1 [0; 2] vs. 0.5 [0; 2]; Z = −1.82; *p* = 0.07). Nausea and vomiting (one or two episodes) were observed mainly on the day of surgery and postoperative day 1. There were no significant differences between groups in the administration of antiemetic drugs (*p* > 0.05). Other side effects, such as urinary retention, pruritis, bradycardia, and dizziness, also presented no significant differences between the intravenous and epidural groups (*p* > 0.05).

This study reported no cases of oversedation. The only incidence of a sedation score of 3 (patient asleep, easy to wake) was noted on the day of surgery, with a similar frequency in the intravenous and epidural groups (*n* = 26, 46.4% vs. *n* = 13, 32.5%; χ^2^ = 1.88; *p* = 0.17).

Respiratory depression, requiring the use of naloxone, occurred in four patients from the intravenous group and two patients from the epidural group immediately upon admission to the intensive care unit (Steward scores < 5/6). Therefore, we did not consider respiratory depression as a side effect of postoperative analgesia.

Harmless neurological complications were documented in patients treated by the epidural route. Numbness and sensory disturbances in the upper limbs were reported by seven patients (vanished spontaneously), while anisocoria and Horner’s syndrome were documented in five and three patients, respectively (resolved after decreasing the infusion of ropivacaine with fentanyl). The occurrence of the above complications did not prevent the continuation of epidural analgesia.

### 3.6. Patient Satisfaction

The mean average satisfaction scores on the day of surgery and consecutive postoperative days and total satisfaction scores were similar in both groups (*p* > 0.05). Patients were very satisfied with their postoperative pain management (the median was 9.7 [8.8; 10]) on postoperative days 0–3. The minimum satisfaction score was 5/10 in the epidural group and 7/10 in the intravenous group. In the postoperative period, 50% (on the day of surgery) to 61% of the patients (on postoperative day 2) were completely satisfied (NRS = 10).

## 4. Discussion

The results of this study suggest that the intravenous infusion of morphine as a component of a multimodal pain management protocol for patients undergoing the Ravitch procedure may offer superior postoperative analgesia than the thoracic epidural infusion of ropivacaine with fentanyl, and this was associated with a lower need for additional analgesics. The profile of anxiety, side effects, and patient satisfaction was similar in both groups.

Our institutional management of postoperative pain in pediatric patients undergoing thoracic surgery has changed over time-the 7th edition of the multimodal analgesic protocol is currently in force. Historically, pain relief in this patient population relied on continuous intravenous morphine. The epidural technique was incorporated into our clinical practice in 2006. It gained popularity and became the analgesic method of choice during the postoperative period. However, the results obtained in this study surprised us, as we expected to find a better analgesic effect of epidural analgesia than conventional analgesia.

We assumed, based on previous evidence [26], that pain management by the epidural route is the most effective method for postoperative pain control in children and adolescents with malformations of the thoracic wall. The recent systematic review and meta-analysis performed by Heo et al. [15] also revealed, contrary to our study, that intravenous analgesia had a worse analgesic effect than epidural analgesia. On the other hand, Man et al. [17] found in their retrospective study that implementation of a standardized, comprehensive multimodal analgesic protocol (intravenous hydromorphone plus intravenous paracetamol/diazepam/ketorolac plus oral gabapentin) can be as effective as a multimodal approach containing epidural ropivacaine with clonidine (plus intravenous morphine or hydromorphone or ketorolac). It should be noted that the above-mentioned studies were carried out among pediatric patients undergoing minimally invasive repair of the pectus excavatum as described by Nuss [27]. The Nuss and Ravitch procedures are similar with respect to the incidence of general complications or the length of hospital stay [28], but there are no clear results indicating which procedure is more painful [9,11].

We also found that patients treated with intravenous morphine required significantly fewer doses of metamizole than patients treated with a thoracic epidural infusion. This result seems to be an additional indicator confirming the superiority of intravenous analgesia with morphine. Metamizole has been introduced into our clinical practice as a rescue drug because, besides its excellent analgesic properties, it exhibits synergism with non-steroidal anti-inflammatory drugs, paracetamol, and opioids [29]. To minimize the risk of agranulocytosis, we have used this medicinal product for a short time [30]. The surveys conducted by Fieler et al. [31] and Witschi et al. [32] confirmed that metamizole is also frequently used by German anesthesiologists in children in the perioperative setting and cases of serious adverse drug reactions such as hemodynamic, anaphylactic, and respiratory reactions (0.3%) or agranulocytosis are very rare (0.14%).

This study showed that our patients, irrespective of the assignment to the group, had lower pain scores (median pain < 1/10) than patients after Ravitch repair in a study conducted by Mangat et al. (median discharge pain ranged from 1.5/10 to 4.5/10) [11] or Pilkington et al. (median pain 4/10) [14]. This discrepancy might derive from the fact that we defined 2 as the maximum pain score above which pain relief was offered, while the intervention threshold in the study by Mangat et al. [11] was probably 4 (the pain was satisfactorily controlled when pain scores were 4 or less on a scale of 0–10).

Excellent pain relief was reflected in the high satisfaction of all our patients. Schwenkglenks et al. [33] reported that one of the predictors of patient satisfaction in adults was not only the actual pain experience but also the possibility of participation in pain treatment decisions and the patient’s impression of the improvement and appropriateness of care. Espinel et al. [34], in their systematic review related to pediatric patient satisfaction, found positive correlations between satisfaction and increased child knowledge about the operative process. It is worth emphasizing that the nurses at our institution conducted preoperative education individually face-to-face, explaining the principles of perioperative nursing care, pain assessment using the three-step method, and modification of analgesia. Thanks to this knowledge, our patients had a positive attitude towards postoperative pain, and this could have an influence on the number of painkillers administered.

Analgesia provided by a nurse was not only effective but also safe. We did not find any serious complications. Postoperative nausea and vomiting (PONV) were most common—about 40% of the patients had at least one incident of this side effect up to postoperative day 2, despite antiemetic prophylaxis. Our patients had a high risk of PONV because they were older than 3 years, received opioids, and underwent surgery that lasted longer than 30 min [35,36]. The type of surgery is also known to influence the rate of PONV. Howard et al. [36] noted that pediatric patients undergoing cardiothoracic surgery more frequently suffered from PONV than patients after general, plastic, or urology surgery.

In patients treated via the thoracic epidural route, we observed harmless and very rare reported in medical literature neurological complications, such as anisocoria and Horner’s syndrome [37,38,39]. A plausible explanation of the mechanism of this complication is the cephalad spread of the local anesthetic solution [38]. Researchers reported that these adverse events may resolve themselves several hours after reducing the local anesthetic infusion or when stopping the infusion [37,40].

### Limitations

The study was randomized and blinded, but not to compare the efficacy of intravenous and epidural analgesia but to assess whether perioperative gabapentin use has an impact on the intensity of the pain (primary outcome) in each study arm. 

The estimated test power indicates that the sample size was sufficient to show that the differences between the intravenous vs. epidural group are statistically significant (the significance level of α = 0.05) only in the case of average/maximum pain at rest (power: 0.89/0.95) and maximum pain when deep breathing (power: 0.85). The test power is low in the case of average pain during deep breathing and average/maximum pain during coughing (0.36, 0.67, and 0.78, respectively).

The lack of a standardized tool to assess satisfaction with postoperative analgesia is also a limitation of the study, as is the fact that the cohorts come from one treatment facility (a single-center cohort study).

## 5. Conclusions

Continuous intravenous morphine may provide superior pain control but was comparable with a thoracic epidural infusion of ropivacaine with fentanyl for secondary outcomes. Randomized studies are needed to support this conclusion.

## 6. Practical Implications of the Study

The results of pain intensity and the adverse events profile suggest that both intravenous morphine and epidural ropivacaine/fentanyl, the main components of multimodal analgesia, are effective and safe, provided that patients’ health condition is closely monitored and analgesia is modified by nurses. Differences in pain intensity at such a low level (average NRS < 1/10) seem to be clinically insignificant. However, the validity of using this invasive method in our clinical practice should be considered due to the potential risk of serious adverse events related to the insertion of an epidural catheter, its maintenance, and the administration of local anesthetics. We plan to conduct a randomized clinical trial to definitively determine whether intravenous analgesia provides better pain control than epidural analgesia. 

## Figures and Tables

**Figure 1 ijerph-19-11291-f001:**
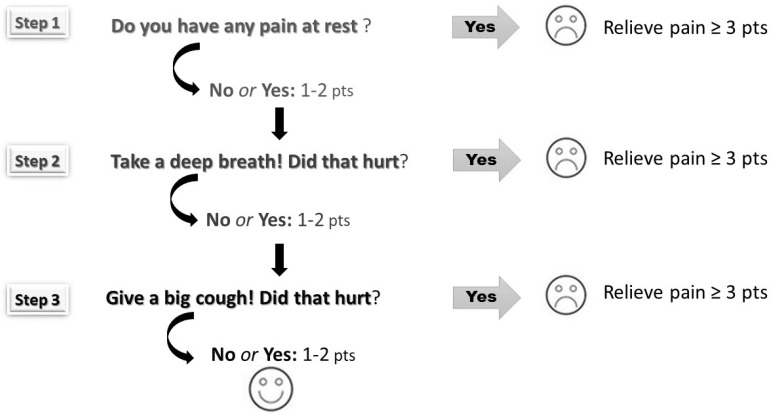
The three-step pain assessment method (the pain was assessed at rest, during deep breathing, and coughing according to the Numerical Rating Scale ranging from 0 points to 10 points).

**Figure 2 ijerph-19-11291-f002:**
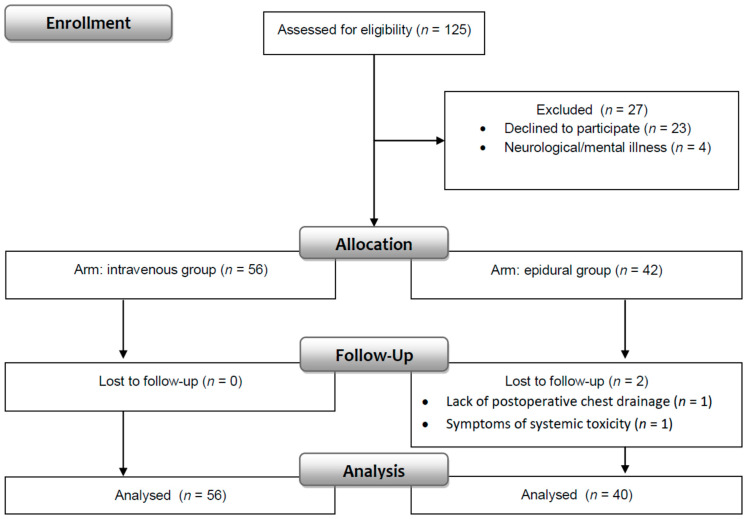
Consort flow diagram.

**Table 1 ijerph-19-11291-t001:** Protocol for the perioperative and intraoperative administration of drugs in the intravenous and epidural groups.

Procedure	Name of Drug, Route, and Dose	
Premedication	midazolam hydrochloride (p.o.) 0.2–0.5 mg/kg	
Antiemetic prophylaxis	ondansetron hydrochloride (i.v.) 0.1 mg/kg up to 4 mg	
Pre-emptive analgesia	paracetamol (i.v.) 15 mg/kg;	
	^1^ non-steroidal anti-inflammatory drug: ibuprofen (p.r.) 10 mg/kg or ketoprofen (i.v.) 1 mg/kg	
Induction of anesthesia	fentanyl (i.v.) 1–5 μg/kg	
	propofol (i.v.) 3–5 mg/kg	
	^2^ rocuronium bromide (i.v.) 1 mg/kg or pancuronium bromide (i.v) 0.1 mg/kg	
Maintenance of anesthesia	desfluranum 8–10 vol% in an oxygen/air mixture	
Intraoperative analgesia		
–intravenous group	fentanyl (i.v.) every 20–30 min; 1–5 μg/kg	
–epidural group	2% lidocaine 2 mg/kg	
	*After 15 min*	
	1% ropivacaine 1–3 mg/kg	
	*After 60 min*	
	0.2% ropivacaine with fentanyl 5 µg/mL; 0.1 mL/kg/per hour	
Postoperative analgesia		
–intravenous group	morphine 0.02–0.06 mg/kg per hour	
–epidural group	0.2% ropivacaine with fentanyl 5 µg/mL; 0.1 mL/kg per hour	
–both groups	paracetamol (i.v.) every 6 h; maximum of 60 mg/kg daily	
	ibuprofen (p.r.) every 8 h 10 mg/kg; up to 30 mg/kg daily or ketoprofen (i.v.) every 8 h 1 mg/kg; maximum of 200 mg daily	
	metamizole (rescue drug) 20 mg/kg; maximum of 2.5 g daily	
	^3^ tramadol (i.v.) 1–2 mg/kg; maximum of 400 mg daily	
Antiemetic treatment	^4^ ondansetron hydrochloride (i.v.) every 8 h up to the second postoperative day; 0,1 mg/kg up to 4 mg	
	^5^ metoclopramide hydrochloride (i.v.) 0.1–0.2 mg/kg	
	^6^ dexamethasone (i.v.) 0.15 mg/kg up to 5 mg	

^1^ ibuprofen was given to children <14 years of age; ^2^ rocuronium was given to children up to 10 years of age; ^3^ tramadol was given on the third postoperative day when the infusion of morphine and ropivacaine with fentanyl was stopped due to removal of the chest drainage; ^4,5^ not to be combined with tramadol; ^5,6^ used according to physician’s decision when ondansetron hydrochloride failed.

**Table 2 ijerph-19-11291-t002:** Demographic and clinical data in the intravenous and epidural groups.

Variable		Intravenous	Epidural	*p* Value
		*n* = 56	*n* = 40	
Age (years)		14 [13; 16]	14 [12; 15]	0.20
Body height (cm)		171 [164; 177]	168 [158; 176]	0.36
Body weight (kg)		54 [45; 60]	52 [44; 58]	0.14
BMI		18 [17; 20]	17 [16; 19]	0.08
Gender	Female	6 (10.7)	6 (15.0)	0.54
	Male	50 (89.3)	34 (85.0)	
ASA	1	52 (92.9)	27 (67.5)	0.002
	2	4 (7.1)	13 (32.5)	
Perioperative gabapentin		28 (50)	20 (50)	1.00
Before induction	Heart rate (beat min^−1^)	86 [77; 94]	95 [86; 108]	0.001
	Systolic blood pressure (mmHg)	117 [105; 126]	105 [94; 120]	0.0002
	Diastolic blood pressure (mmHg)	70 [60; 76]	60 [50; 70]	0.018
	Oxygen saturation (%)	98 [98; 99]	99 [98; 99]	0.06
Duration of anesthesia (min)		195 [178; 209]	195 [167; 212]	0.72
Duration of surgery (min)		140 [126; 153]	120 [105; 142]	0.004
Duration of extubating (min)		15 [10; 20]	15 [10; 20]	0.40

Results presented as median, lower and upper quartile or absolute number (percentage); ASA—American Society of Anesthesiologists physical status.

**Table 3 ijerph-19-11291-t003:** Average and maximal postoperative pain in the intravenous and epidural groups.

Pain	Intravenous	Epidural	*Z*	*p* Value	Effect Sizes
(Postoperative Days 0–3)	*n* = 56	*n* = 40			
	Average pain				
At rest	0.2 [0.1; 0.4]	0.4 [0.2; 0.9]	−2.97	0.003	0.6
During deep breathing	0.2 [0.1; 0.3]	0.3 [0.1; 0.4]	−2.27	0.023	0.5
During coughing	0.2 [0.1; 0.4]	0.4 [0.3; 0.5]	−2.43	0.015	0.5
Maximal pain					
At rest	1.2 [0.7; 2.2]	2.0 [1.0; 3.5]	−3.29	0.001	0.7
During deep breathing	0.7 [0.5; 1.2]	1.2 [0.7; 2.0]	−2.85	0.004	0.6
During coughing	1.0 [0.5; 1.5]	1.5 [1.0; 1.7]	−2.98	0.003	0.6

Results of pain according to the NRS scale (range 0–10) are presented as median, lower and upper quartile; Effect Sizes are shown as Cohen’s d.

**Table 4 ijerph-19-11291-t004:** Trait and state anxiety in the intravenous and epidural groups.

Anxiety	Intravenous	Epidural	*p* Value
	*n* = 56	*n* = 40	
Trait (sten)	5 [3; 6]	5 [4; 6]	0.12
State preoperative (sten)	6 [5; 7]	7 [6; 8]	0.06
State postoperative (sten)	5 [4; 6]	6 [4; 7]	0.07

Results are presented as median, lower and upper quartile; sten: range 1–10.

**Table 5 ijerph-19-11291-t005:** Side effects of postoperative analgesia.

Variable	Intravenous	Epidural	*p* Value
	*n* = 56	*n* = 40	
Nausea and vomiting	28 (50.0)	28 (70.0)	0.05
Urinary retention-bladder catheterization	3 (5.4)	0 (0.0)	0.26
Pruritus	2 (3.6)	3 (7.5)	0.64
Bradycardia	2 (3.6)	4 (10.0)	0.23
Dizziness	1 (1.8)	1 (2.5)	1.00

Results presented as absolute numbers and percentages; Occurrence of nausea/vomiting: 39.6% (*n* = 38)—on the day of surgery, 34.4% (*n* = 33)—on postoperative day 1, 12.5% (*n* = 12)—on postoperative day 2, 10.5% (*n* = 10)—on postoperative day 3. Total incidence of vomiting: 1 incidence (*n* = 26), 2 incidences (*n* = 14), 3 incidences (*n* = 5), 4 incidences (*n* = 4), 5 incidences (*n* = 4), 6 incidences (*n* = 1), 7 incidences (*n* = 2).

## Data Availability

A dataset will be made available upon request to the corresponding authors one year after the publication of this study. The request must include a statistical analysis plan.

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
