# Peer review of "Intravenous Morphine Infusion versus Thoracic Epidural Infusion of Ropivacaine with Fentanyl after the Ravitch Procedure—A Single-Center Cohort Study"

_ijerph, 2022, doi:10.3390/ijerph191811291_

Round 1
Reviewer 1 Report
line 85: clinicaltrials.gov is spelled incorrectly
line 137: change "A" to "All", or remove "A" entirely
line 138: need a period at the end of the sentence
line 158: change "help" to "helps"
line 168: change "questioners" to "questionnaires"
line 184: change "inadequately" to "inadequate"
line 280: change "naloxon" to "naloxone"
line 300: there is an opening parenthesis, but no closing parenthesis
line 321: change "adolescent" to "adolescents"
line 404: change "definitely" to "definitively"
---
Since there is variation in which thoracic space the epidural catheter was placed for patients in the epidural arm, I would suggest a brief explanation that all of them, regardless of which thoracic space accessed for each patient in the study, would cover the required sensory level for this procedure. It may seem obvious for those who do this frequently, but it may not be obvious for readers of the journal who don't do it frequently.
Author Response
Dear Reviewer,
We would like to thank you for a review of our manuscript and all of your valuable remarks. We have addressed them all in detail below
With kind regards,
Authors
Point 1. Since there is variation in which thoracic space the epidural catheter was placed for patients in the epidural arm, I would suggest a brief explanation that all of them, regardless of which thoracic space accessed for each patient in the study, would cover the required sensory level for this procedure. It may seem obvious for those who do this frequently, but it may not be obvious for readers of the journal who don't do it frequently.
Response 1. In the revised Methods (2.3.2. Epidural catheter), we have added ‘’a T4-T7 sensory level is appropriate for Ravitch procedure’’.
Point 2.
line 85: clinicaltrials.gov is spelled incorrectly
line 137: change "A" to "All", or remove "A" entirely
line 138: need a period at the end of the sentence
line 158: change "help" to "helps"
line 168: change "questioners" to "questionnaires"
line 184: change "inadequately" to "inadequate"
line 280: change "naloxon" to "naloxone"
line 300: there is an opening parenthesis, but no closing parenthesis
line 321: change "adolescent" to "adolescents"
line 404: change "definitely" to "definitively"
Response 2. In the revised manuscript all words have written correctly as suggested.
Reviewer 2 Report
This is a very interesting study.
Although there are new techniques for analgesia available today, as you mentioned in your paper, these two that you compared are still the most important at the moment.
Concerns with complications due to epidural analgesia are probably under reported and many thoracic surgeons would prefer not to use this technique in their patients, as myself. So, a search for a valid option is of paramount importance.
The only question I have after reading your nice paper is if there would be similar results with the Nuss procedure instead of the Ravitch operation.
Author Response
Dear Reviewer,
We would like to thank you for a detailed review of our manuscript and all of your valuable remarks. All changes in the manuscript appear in red font and we have addressed them all in detail below.
With kind regards,
Authors
Point 1. This is a very interesting study. Although there are new techniques for analgesia available today, as you mentioned in your paper, these two that you compared are still the most important at the moment. Concerns with complications due to epidural analgesia are probably under reported and many thoracic surgeons would prefer not to use this technique in their patients, as myself. So, a search for a valid option is of paramount importance. The only question I have after reading your nice paper is if there would be similar results with the Nuss procedure instead of the Ravitch operation.
Response. In our institution, patients are not operated using the technique described by Nuss. However, we suppose that applying the same analgesia to these patients as in our study would have produced comparable analgesic effects. We believe that adequate pain relief can be achieved not only with an appropriate analgesic method, but also with regular pain measurements using the three-step method (at rest, during deep breathing and during coughing), and appropriate supervision of pain relief procedures in clinical practice. It is worth emphasizing that nurses in our institution are obliged to relieve pain> 2/10.
Reviewer 3 Report
I read the work “Intravenous morphine infusion versus thoracic epidural infusion of ropivacaine with fentanyl after the Ravitch procedure - a single-center cohort study” with pleasure.
I congratulate with the authors for the interesting topic and the good number of patients enrolled.
The secondary outcome (the number of doses of metamizole during postoperative analgesia) in addition to the primary outcome (the measurement of NRS) increases the quality of the study. The NRS can be administered verbally or in a written format, is simple and easily understood and scored, but the principal weakness of the NRS is that, statistically, it does not have ratio qualities.
However, I have one minor comments to add:
- In the table 2 it is shown that 52 (92.9) patients underwent IV anesthesia had an ASA 1 and 27 (67.5) patients who underwent epidural anesthesia had ASA 1, the difference was statistically significant (p=0.002). Could you explain if ASA classification could be a confounding factor in the comparison of NRS of pain between the two groups? If yes, you should consider it in the paragraph “Limitations” in the manuscript.
The manuscript is well-written, the methodology is of quality, and the findings contribute to the scientific knowledge database. I therefore recommend acceptance.
Author Response
Dear Reviewer,
We would like to thank you for a detailed review of our manuscript and all of your valuable remarks. All changes in the manuscript appear in red font and we have addressed them all in detail below.
With kind regards,
Authors
I read the work “Intravenous morphine infusion versus thoracic epidural infusion of ropivacaine with fentanyl after the Ravitch procedure - a single-center cohort study” with pleasure. I congratulate with the authors for the interesting topic and the good number of patients enrolled. The secondary outcome (the number of doses of metamizole during postoperative analgesia) in addition to the primary outcome (the measurement of NRS) increases the quality of the study. The NRS can be administered verbally or in a written format, is simple and easily understood and scored, but the principal weakness of the NRS is that, statistically, it does not have ratio qualities.
However, I have one minor comments to add:
- In the table 2 it is shown that 52 (92.9) patients underwent IV anesthesia had an ASA 1 and 27 (67.5) patients who underwent epidural anesthesia had ASA 1, the difference was statistically significant (p=0.002). Could you explain if ASA classification could be a confounding factor in the comparison of NRS of pain between the two groups? If yes, you should consider it in the paragraph “Limitations” in the manuscript.
The manuscript is well-written, the methodology is of quality, and the findings contribute to the scientific knowledge database. I therefore recommend acceptance.
Response. The Mann Whitney test results showed no significant differences between ASA and pain (p > 0.05)